# Single-Walled Carbon Nanotubes as Enhancing Substrates for PNA-Based Amperometric Genosensors

**DOI:** 10.3390/s19030588

**Published:** 2019-01-30

**Authors:** Simone Fortunati, Andrea Rozzi, Federica Curti, Marco Giannetto, Roberto Corradini, Maria Careri

**Affiliations:** Dipartimento di Scienze Chimiche, della Vita e della Sostenibilità Ambientale, Università di Parma, Parco Area delle Scienze 17/A, 43124 Parma, Italy; simone.fortunati@studenti.unipr.it (S.F.); andrea.rozzi@studenti.unipr.it (A.R.); federica.curti@studenti.unipr.it (F.C.); roberto.corradini@unipr.it (R.C.)

**Keywords:** PNA-based genosensors, single-walled carbon nanotubes, glassy carbon

## Abstract

A new amperometric sandwich-format genosensor has been implemented on single-walled carbon nanotubes screen printed electrodes (SWCNT-SPEs) and compared in terms of performance with analogous genoassays developed using the same methodology on non-nanostructured glassy carbon platforms (GC-SPE). The working principle of the genosensors is based on the covalent immobilization of Peptide Nucleic Acid (PNA) capture probes (CP) on the electrode surface, carried out through the carboxylic functions present on SWCNT-SPEs (carboxylated SWCNT) or electrochemically induced on GC-SPEs. The sequence of the CP was complementary to a 20-mer portion of the target DNA; a second biotin-tagged PNA signalling probe (SP), with sequence complementary to a different contiguous portion of the target DNA, was used to obtain a sandwich hybrid with an Alkaline Phosphatase-streptavidin conjugate (ALP-Strp). Comparison of the responses obtained from the SWCNT-SPEs with those produced from the non-nanostructured substrates evidenced the remarkable enhancement effect given by the nanostructured electrode platforms, achieved both in terms of loading capability of PNA probes and amplification of the electron transfer phenomena exploited for the signal transduction, giving rise to more than four-fold higher sensitivity when using SWCNT-SPEs. The nanostructured substrate allowed to reach limit of detection (LOD) of 71 pM and limit of quantitation (LOQ) of 256 pM, while the corresponding values obtained with GC-SPEs were 430 pM and 1.43 nM, respectively.

## 1. Introduction

In recent decades, several novel materials with promising properties for electrochemical sensing have been developed. In particular, nanomaterials such as nanowires, nanoparticles, nanosheets and nanotubes have been extensively used for electrode fabrication to achieve a considerable improvement of the overall performance, as reported in recent review papers dealing with the discussion of nanomaterial-based electrochemical sensor and biosensor platforms for biological and biomedical applications [1] and of the latest trends in emerging nanostructured electrochemical biosensors [2]. Among these, Carbon NanoTubes (CNTs) have received increasing attention and are nowadays widely used in the field of biosensing [3,4,5,6,7,8,9,10,11]. These structures are arranged as one or more graphene sheets rolled-up to form a hollow tube of diameter ranging from 0.4 to 2 nm. CNTs can be divided into two classes: Single-Walled (SWCNT) consisting of a single tubular unit and Multi-Walled Carbon NanoTubes (MWCNT), which are formed by multiple coaxial graphene layers. Grafting of such nanostructures on electrode substrates can radically lead to signal enhancement due to an increase in the electroactive surface area. Furthermore, CNTs exhibit an enhancement of the electron transfer phenomena due to their high electronic conductivity and resulting outstanding electrical performance, and thus can be used to develop electrochemical biosensors, such as genosensing devices.

Genosensors represent a very promising analytical tool and a leading approach due to their specificity, speed, cost-effectiveness and remarkably wide application field, ranging from medical diagnostic applications [12,13,14] to food safety and authenticity assessment [15,16,17,18,19]. These systems rely on the highly specific and efficient recognition mechanism between nucleobases of oligonucleotide strands, the wide majority of assay involving a hybridization event between an immobilized Capture Probe (**CP**) and a target DNA or RNA in solution. Electrochemical genosensors translate the hybridization event into measurable electrochemical signals using various strategies, e.g., labelling of the target DNA, use of a labelled probe, etc. In our approach, by using a probe labelled with biotin (Signalling Probe; **SP**), complementary to a portion of the target DNA flanking the **CP**-binding sequence, a **CP**/Target/**SP** sandwich complex is obtained. A 45-mer sequence specific of genetically modified soy (*Roundup Ready* soybean) was selected as target DNA. The electrochemical signal is then produced by the interaction between the biotin label and a Streptavidin-Alkaline Phosphatase conjugate (ALP-Strp) in the presence of Hydroquinone Diphosphate (HQDP) as enzymatic substrate; the electroactive Hydroquinone (HQ) product is oxidized by the electrode to quinone, giving a signal proportional to the amount of **SP** hybridized on the electrode surface, and hence to the amount of target DNA. **CP** and **SP** can be either oligonucleotidic (DNA or RNA) or DNA mimics such as Peptide Nucleic Acids (PNAs) or Locked Nucleic Acids (LNAs). PNAs [20] are extremely good structural mimics of DNA with a 2-aminoethylglycine backbone, and PNA oligomers are able to form very stable duplexes with Watson-Crick complementary DNA or RNA targets, giving rise to the hybridization event necessary for sensing. The higher stability of these duplexes compared to that of double-stranded DNA allows to use short sequences with still high response, thus reducing the possible sequences interfering by partial hybridization with the CP and SP probes. Furthermore, the well-established solid-phase synthesis of PNAs makes them easy to produce and to modify to tune their properties [21].

In the frame of a research program aimed at the development of novel amperometric PNA-based genosensors, in this work a genosensor assay with the above-described sandwich format was devised to perform a comparison between glassy carbon- (GC-SPEs) and SWCNT-modified screen printed electrodes (SWCNT-SPEs) modified through covalent immobilization of PNA capture probes. Findings of this study evidenced the enhancement in the performance of PNA-based genosensing devices through implementation on CNTs, if compared with the corresponding devices implemented on non-nanostructured glassy carbon electrodes. The validation of genosensor based on CNTs for the determination of non-amplified genomic DNA extracted from flour real samples of genetically modified soybean was discussed in another of our recent papers [22].

## 2. Materials and Methods

### 2.1. Chemicals

*N*,*N*-Diisopropylethylamine (DIPEA), piperidine, *N*,*N*,*N*′,*N*′-Tetramethyl-O-(1H-benzotriazol-1-yl)uronium hexafluorophosphate (HBTU), acetic anhydride, trifluoroacetic acid (TFA), m-cresol, biotin, Fmoc-glycine, *N*^α^-Fmoc-*N*^ε^-Boc-L-lysine, Rink Amide resin, *N*-(3-Dimethylaminopropyl)-*N*′-ethylcarbodiimide hydrochloride (EDC), *N*-Hydroxysuccinimide (NHS), 4-morpholineethanesulfonic acid monohydrate (MES), sodium bicarbonate (NaHCO_3_), sodium dodecyl sulfate (SDS), disodium hydrogen phosphate (Na_2_HPO_4_), Trizma^®^ base, ethylendiaminetetraacetic acid disodium salt dihydrate (Na_2_EDTA), sodium chloride (NaCl), albumin from bovine serum (BSA), magnesium chloride anhydrous (MgCl_2_), streptavidin−alkaline phosphatase from *Streptomyces avidinii* (ALP-Strp), Tween^®^ 20 and sulphuric acid (H_2_SO_4_) were purchased from Sigma-Aldrich (Milan, Italy). Solvents for peptide synthesis and purification were purchased from Sigma-Aldrich (Milan, Italy) and used without further purification, except for *N*,*N*-Dimethylformamide (DMF) dried over 4 Å molecular sieves and purged with nitrogen to avoid the presence of dimethylamine. Hydroquinone diphosphate (HQDP) was from Metrohm Italiana (DropSens DRP-HQDP) (Origgio, Italy). PNA monomers and spacers AEEA (2-[2-(Fmoc-amino)ethoxy]ethoxyacetic acid) were from LGC LINK (Teddington, UK).

Synthetic DNA probes were purchased from biomers.net GmbH (Ulm, Germany) with the following sequences: *Target DNA*: 5′-ACT TGG GGT TTA TGG AAA TTG GAA TTG GGA TTA AGG GTT TGT ATC-3′.

Double-distilled and deionized water purified with a Milli-Q system was used for the preparation of the buffer solutions.

### 2.2. Preparation of Buffer Solutions

Buffer solutions were prepared according to the following compositions:
“MES buffer”: 0.1 M MES (pH of the solution adjusted to pH 5 with NaOH).Tris buffered saline (TBS): 0.1 M Trizma^®^ base, 0.02 M MgCl_2_ (pH of the solution adjusted to pH 7.4 with HCl).Tris buffered saline-Tween (TBS-T): 0.1 M Trizma^®^ base, 0.02 M MgCl_2_, 0.05% w/v Tween 20^®^ (pH of the solution adjusted to pH 7.4 with HCl).“Carbonate buffer” (CB): 0.1 M NaHCO_3_, 0.1% w/v SDS (pH of the solution adjusted to pH 9 with NaOH).“Hybridization buffer”: 0.3 M NaCl, 0.02 M Na_2_HPO_4_, 0.1 mM EDTA (pH of the solution adjusted to pH 7.4 with HCl).“Blocking Buffer” (BB): 20 mg mL^−1^ BSA in TBS (pH 7.4).“Reading buffer” (RB): 0.1 M Trizma^®^ base, 0.02 M MgCl_2_ (pH of the solution adjusted to pH 9.8 with HCl).

### 2.3. PNA Synthesis and Characterization

PNA **CP** (H-AEEA-AEEA-GAT ACA AAC CCT TAA TCC CA-Gly-NH_2_) and **SP** (**SP**: Biotin-AEEA-AEEA-AAT TTC CAT AAA CCC CAA GT-Lys(NH_3_^+^)-NH_2_) were synthesized by the automatic synthesizer Biotage Syro I in 2.5 mL polypropylene reactors, using the procedures described elsewhere [23].

PNA oligomers were purified by RP-HPLC using a XTerra^®^ Prep RP18 column (7.8 x 300 mm, 10 μm) (Waters Corporation, Milford, MA, USA). HPLC conditions: 5.00 min in water 0.1% TFA, then linear gradient from water 0.1% TFA to 50% acetonitrile 0.1 % TFA in 30 min at a flow rate of 4.0 mL min^−1^, and then characterized using UPLC-MS analysis on a Waters Acquity Ultra Performance LC equipped with Waters Acquity SQ Detector and electrospray interface.

PNA concentrations were determined by UV absorption at 260 nm using a Lambda BIO 20 Perkin Elmer Spectrophotometer (Perkin Elmer, San Antonio, TX, USA) and calculated from the following extinction coefficients of the nucleobases: Adenine 13700 l mol^−1^ cm^−1^, Cytosine 6600 l mol^−1^ cm^−1^, Guanine 11700 l mol^−1^ cm^−1^, Thymine 8600 l mol^−1^ cm^−1^.

### 2.4. Genosensor Setup and Method Validation

Genosensors were assembled on SWCNT-SPEs and GC-SPEs purchased by Metrohm Italiana (DropSens DRP-110SWCNT and DRP-110, respectively) (Origgio, Italy).

All electrochemical measurements were performed with a PGSTAT-204 potentiostat/galvanostat produced by Metrohm Italiana, equipped with NOVA 2.1.3 Advanced Electrochemical Software and connected to DropSens DRP-DSC plug.

All the measurements carried out to assess the best experimental conditions in terms of proper concentration of **CP** and **SP,** as well as the signal values for construction of calibration curves, were replicated three times, carrying out independent measurements on different SPEs. Mean values and standard deviation are shown in all figures.

Method validation was performed by assessing the linearity range and calculating Limit of Detection (LOD) and Limit of Quantification (LOQ) according to “Eurachem Guides” (http://www.eurachem.org/index.php/ publications/guides/mv). Precision was evaluated in terms of intermediate precision by carrying out three independent assays for each target DNA concentration level.

#### 2.4.1. Preparation of Modified Glassy Carbon Electrode

Modification of glassy carbon electrode by electrochemical oxidation was performed by casting 50 µL of 0.1 M H_2_SO_4_ on the electrode surface and scanning the potential between −1 and +1.7 V by Cyclic Voltammetry (10 scans, scan rate=100 mVs^−1^; step potential= +0.00305 mV) [24]. After treatment, SPEs were rinsed thoroughly with water.

As for SWCNT-SPEs, immobilization of **CP** was performed directly without any treatment of the electrode surface, as reported in Section 2.4.2.

#### 2.4.2. Capture Probe Immobilization

The carboxylic functions on the SWCNT-SPEs and on the GC-SPEs were activated by incubation of 50 µL of 0.2 M EDC and 0.05 M NHS in MES buffer for 1 h at room temperature. After removal of the solution by rinsing with water, 50 µL of 500 nM **CP** in carbonate buffer was incubated for 2 h at room temperature, after which unreacted species were removed by rinsing with water. In order to prevent non-specific interaction of probes with the electrode substrates [25], a blocking step was performed by depositing 50 µL of 500 nM pyrene in DMSO. The SPEs surface was then washed with DMSO followed by water.

#### 2.4.3. Hybridization of Target DNA and Signaling Probe in Homogeneous Phase

Properly diluted solutions of target DNA and **SP** in Hybridization Buffer were mixed together in order to reach a final concentration of 20 nM **SP** and the desired target DNA concentration. The mixture was left under agitation at 1000 rpm for 3 h at room temperature. This solution was subsequently transferred on the electrode surface and incubated for 2h. The SPEs were then rinsed with 0.05% Tween followed by water.

#### 2.4.4. Enzymatic Labelling and Reading of the Electrochemical Genoassay

The ALP-Strp conjugate was 100-fold diluted in BB and incubated on the SPEs surface for 15 min at room temperature before washing with TBS-T followed by TBS. Electrochemical read-out was performed by Differential Pulse Voltammetry (DPV) measurements using 50 µL of a 1 mg mL^−1^ solution of HQDP dissolved in RB, which was left in contact with the sensor surface for a fixed time of 2 min 30 s immediately prior to measurement. The incubation time of ALP-Strp was set at 15 min since longer times (30 and 45 min) did not result in statistically significant improvements (*p* > 0.05) in the response. An analogous criterion was applied for the assessment of the proper incubation time of HQDP. DPV curves were acquired by scanning the potential between −0.5 V and +0.3 V (step potential= +0.00495 V, modulation amplitude= +0.04995 V, modulation time= 0.102 s, interval time= 0.4 s) and recording the signal assigned to the oxidation of HQ, generated by ALP-promoted enzymatic dephosphorylation of HQDP, to Quinone (Q). The peak current is associated with the amount of HQ produced, which in turn is related to the amount of the **CP**/Target/**SP** sandwich formed. At least three replicate measurements were carried out for all DNA concentration levels.

## 3. Results and Discussion

The working principle of the developed genosensor is summarized in Scheme 1: a PNA **CP** bearing an amino function was covalently bound to the electrode surface through the carboxylic groups present on SWCNT-SPEs or electrochemically formed on GC-SPEs.

The sequence of the **CP** was complementary to a 20-mer portion of the target DNA; a second biotin-tagged PNA-**SP**, with sequence complementary to a different contiguous portion of the target DNA, was used to obtain a sandwich hybrid [16,26] with an ALP-Strp conjugate.

The read-out of the electrochemical genosensor assay is carried out using HQDP as enzymatic substrate, which is enzymatically converted to HQ, yielding a voltammetric signal proportional to the amount of PNA-**SP** hybridized on the electrode surface.

The protocol of the developed genoassay consisted of a first versatile hybridization of DNA with **SP** carried out in homogeneous phase in a disposable plastic tube; this hybrid is subsequently transferred on the **CP**-functionalized SPEs. The final step consisted of the incubation of the enzyme-conjugate ALP-Strp reacting with the biotin tag of **SP**, followed by drop-casting of HQDP on the electrode surface in order to cause dephosphorylation leading to the analytical signal acquired by DPV.

### 3.1. Genosensor Setup

The PNAs necessary as **CP** and **SP** in Scheme 1 were obtained by automatic peptide synthesis on Chemmatrix Rink Amide resin; the PNA sequence was conjugated to an AEEA spacer for **CP** and to biotin for **SP**. The sequence of the PNA probes and the positioning with respect to the target DNA sequence is depicted in Scheme 2.

In order to find the best measuring conditions, the proper concentration of **CP** and **SP** was assessed for both electrode substrates. As for **CP**, to guarantee an exhaustive coverage of the sensing surface, the concentration of the solution used for the immobilization on both SWCNT-SPEs and GC-SPEs was fixed at 500 nM, present in a large excess compared to the other constituents of the sandwich. Conversely, the effect of the **SP** concentration was thoroughly studied in terms of signal-to-background ratio (S/B) achievable using the different electrode substrates.

The **SP** concentration effect was explored over the 10–50 nM range; the highest concentration tested was chosen to evaluate the extent of non-specific adsorption phenomena that should increase with **SP** concentration. A progressively decreasing trend was evidenced at lower **SP** concentrations (Figure 1), allowing us to assess the best concentration for both electrode substrates.

In the case of GC-SPEs, the signal intensity recorded at 20 nM **SP** concentration was too low, a good S/B being achieved only at 50 nM. As for SWCNT-SPEs, a 20 nM **SP** concentration resulted in the best compromise in terms of S/B ratio and in terms of precision associated to dispersion of measured values in the data set.

Figure 2 shows the signals obtained in the presence and absence of target DNA equimolar to SP (50 nM) on both the immobilization substrates; although comparable low background signals were observed, evidencing the efficiency of pyrene as backfilling agent, the presence of target DNA gave a remarkably higher response on the nanostructured SWCNT substrate.

### 3.2. Analytical Performance

SWCNT-SPEs showed higher sensitivity than GC-SPEs (Figure 3), the nanostructured substrate allowing to reach a LOD of 71 pM and a LOQ of 256 pM, while the corresponding values obtained using GC-SPEs were 430 pM and 1.43 nM, respectively.

Exploring linearity of response as a function of DNA concentration for the two electrode substrates, SWCNT-SPEs showed a linear response between 0.25 and 1.75 nM with a more than four-fold higher sensitivity with respect to GC-SPEs, which gave linear response in the 1.5–10 nM range (Figure 3).

These findings assess a better performance exhibited by SWCNT-SPEs compared to GC-SPEs, the slightly higher cost of the carbon nanotube modified screen-printed electrodes being counterbalanced by the improvement in their performance obtained by exploiting the enhancement of electron transfer process offered by carbon nanotubes.

Concerning intermediate precision, good results were achieved for both the electrode substrates (SWCNT-SPE and GC-SPE), relative standard deviations (RSD) being always lower than 10%.

## 4. Conclusions

Findings of this comparative study demonstrate the outstanding enhancement properties of single-walled carbon nanotubes when used as highly efficient and reactive substrates for covalent immobilization of PNA probes. Efficiency of such nanostructures as electrode materials relies in the improvement of the loading capability of the receptors (i.e., PNA **CP**) as well as in the amplification of the electron transfer phenomena involved in the signal transduction mechanism. The typical amplification properties of CNTs for electron transfer phenomena are well known and consolidated, as reported in several studies published in the literature (4–8), and in this work these advantages were confirmed by our findings also in the field of genosensors based on PNA probes. The overall performance of the genosensor based on SWCNT-SPEs proved to be undoubtedly superior to the analogous sensing devices implemented on GC-SPEs in many respects, primarily in terms of sensitivity and benefit-cost ratio, paving the way for the exploitation of these systems for development of smart, efficient and portable tools for diagnostic purposes.

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
