# Peer review of "Single-Walled Carbon Nanotubes as Enhancing Substrates for PNA-Based Amperometric Genosensors"

_sensors, 2019, doi:10.3390/s19030588_

Round 1
Reviewer 1 Report
This article is well written, comprehensive, coherent, and contains valuable information on the Single-walled carbon nanotubes (SWCNTs) as enhancing substrates for peptide nucleic acid-based amperometric genosensors. The authors demonstrated that the overall performance of the genosensor based on SWCNT-screen-printed electrodes (SPEs) proved to be superior to the analogous sensing devices implemented on glassy carbon-SPEs in many respects, primarily in terms of sensitivity and benefit-cost ratio, paving the way to the exploitation of these systems for development of smart, efficient and portable tools for diagnostic purposes. Therefore, the present manuscript deserves publication in a future issue of the Sensors.
Author Response
This article is well written, comprehensive, coherent, and contains valuable information on the Single-walled carbon nanotubes (SWCNTs) as enhancing substrates for peptide nucleic acid-based amperometric genosensors. The authors demonstrated that the overall performance of the genosensor based on SWCNT-screen-printed electrodes (SPEs) proved to be superior to the analogous sensing devices implemented on glassy carbon-SPEs in many respects, primarily in terms of sensitivity and benefit-cost ratio, paving the way to the exploitation of these systems for development of smart, efficient and portable tools for diagnostic purposes. Therefore, the present manuscript deserves publication in a future issue of the Sensors.
Answer R1: We thank the reviewer for appreciating our work and supporting it for publication in Sensors.
Reviewer 2 Report
In this manuscript, the author investigated the single-walled carbon nanotubes screen printed electrode for biological sensing applications. The contents will be suitable for publication from Sensor due to its high quality of discussion however several points should be resived for acceptance due to the low quality of Figure.
p.5, Scheme 1. The resolution of this scheme is quite low for understanding.
Figures 1, 2 and 3. "E-05" should be revised. The unit seems to be mA and "E-05" should be 10-2.
Author Response
In this manuscript, the author investigated the single-walled carbon nanotubes screen printed electrode for biological sensing applications. The contents will be suitable for publication from Sensor due to its high quality of discussion however several points should be resived for acceptance due to the low quality of Figure.
p.5, Scheme 1. The resolution of this scheme is quite low for understanding.
Answer R2.1: We thank the reviewer for the suggestion. The resolution of Scheme 1 has been improved at the highest possible quality, considering the overall size of the image.
Figures 1, 2 and 3. "E-05" should be revised. The unit seems to be mA and "E-05" should be 10-2.
Answer R2.2: We disagree with the Reviewer on this point. The unit of current is mA, thus the E-05 exponent is correct.
Reviewer 3 Report
The manuscript sensors-410230 refere to the SWCNT as substrate for covalent immobilization of PNA probes. The subject and findings are interested, but the authors should improve the paper before publishing.
1. Line 37. Repetition: “The structures are structured …”. Recommendation: “The structures are ordered as…” or “…set-up…” or “…arranges…”.
2. Line 132. Please give more information about “best experimental conditions”…what do you mean. Be more precise.
3. Line 166-171: The paragraph “2.4.3. Hybridization of target DNA …” is doubled.
4. Line 178. The authors should explain why they chose the fixed time of 2 min 30 seconds for contact of the sensor surface with the solution of HQDP dissolved in RB. How did they set this value…please specify.
5. Line 180. Use of “ascribable” is hard to understand for not advanced English speakers, use instead “attributed” or “assigned”.
6. Line 257. “both the devices”- question mark regarding the number of studied sensors. I understand that there are two types studied, SWCNT-SPEs and GC-SPE, but I did not find in the text how many sensor devices structured were tested and compared.
The chapter 3 has to be detailed more. What about the repeatability of the measurements?
7. Line. 264. Conclusion Chapter. Please give more information about your conclusion regarding the “electron transfer phenomena involved in the signal transduction mechanism. How did you reach this conclusion? … since the authors did not provided this kind of data in the manuscript.
General comments:
The paper provides too brief information regarding the sensor preparation. Also, the conclusions are not fully sustained by the experimental work and findings presented in the manuscript body (chapter 3. Results and discussions)
It is recommended a comprehensive review of the proposed paper and the addition of detailed graphs regarding the measurements and solid explanations of detection mechanisms.
Author Response
The manuscript sensors-410230 refere to the SWCNT as substrate for covalent immobilization of PNA probes. The subject and findings are interested, but the authors should improve the paper before publishing.
1. Line 37. Repetition: “The structures are structured …”. Recommendation: “The structures are ordered as…” or “…set-up…” or “…arranges…”.
Answer R3.1: Following the Reviewer’s suggestion, the repetition was avoided replacing the verb.
2. Line 132. Please give more information about “best experimental conditions”…what do you mean. Be more precise.
Answer R3.2: We thank the reviewer for this comment. The best experimental conditions are referred to the proper concentrations of CP and SP, which were assessed performing experiments before acquisition of the calibration lines (see Section 3.1). In the revised manuscript this aspect has been stated more clearly.
3. Line 166-171: The paragraph “2.4.3. Hybridization of target DNA …” is doubled.
Answer R3.3: We apologize for this mistake. The double paragraph has been deleted in the revised manuscript.
4. Line 178. The authors should explain why they chose the fixed time of 2 min 30 seconds for contact of the sensor surface with the solution of HQDP dissolved in RB. How did they set this value…please specify.
Answer R3.4: We thank the reviewer for this relevant comment. A discussion about this study appears in the section 2.4.4 of the revised manuscript (lines 191-193).
5. Line 180. Use of “ascribable” is hard to understand for not advanced English speakers, use instead “attributed” or “assigned”.
Answer R3.5: Following the Reviewer’s suggestion, the term “ascribable” has been replaced with “assigned” in the section 2.4.4 of the revised manuscript.
6. Line 257. “both the devices”- question mark regarding the number of studied sensors. I understand that there are two types studied, SWCNT-SPEs and GC-SPE, but I did not find in the text how many sensor devices structured were tested and compared.
The chapter 3 has to be detailed more. What about the repeatability of the measurements?
Answer R3.6: The statement is referred to the two different electrode substrates (SWCNT-SPEs and GC-SPEs) compared in the study. This aspect has been specified in the section 3.2 of the revised manuscript (line 284), accordingly to the comment. As for the replicated measurement for each experiment, since the sensors are intended for single use, all the obtained mean values and standard deviations were assessed from three independent assays carried out on different SPEs, as specified in line 135 of the revised manuscript.
7. Line. 264. Conclusion Chapter. Please give more information about your conclusion regarding the “electron transfer phenomena involved in the signal transduction mechanism. How did you reach this conclusion? … since the authors did not provided this kind of data in the manuscript.
Answer R3.7: Thank you for pointing this out. The statement was referred to the enhanced response provided by the CNTs with respect to glassy carbon non-nanostructured substrate. The improved sensitivity is explainable considering both the high loading capability of CP on the electrode substrate and the enhancement of the electron transfer processes on which the response of amperometric sensors is based.
The properties of CNTs in terms of amplification of electron transfer phenomena are well known and consolidated, as reported in several studies published in literature, and have been confirmed by our findings also in the field of genosensors based on PNA probes.
General comments: The paper provides too brief information regarding the sensor preparation. Also, the conclusions are not fully sustained by the experimental work and findings presented in the manuscript body (chapter 3. Results and discussions). It is recommended a comprehensive review of the proposed paper and the addition of detailed graphs regarding the measurements and solid explanations of detection mechanisms.
Answer: We disagree with the Reviewer on this comment. In our opinion, the experimental data and the graphs provided are sufficiently exhaustive to describe and rationalize the results of the study. The detection mechanism, clearly described in Scheme 1, is based on the use of biotin as a tag for the enzymatic detection based on the ALP-Strp conjugate, according to a "sandwich" approach, already consolidated and reported in several studies published and reviewed in the literature (see: http://dx.doi.org/10.1016/j.rac.2014.10.006, inserted as new reference (no. 16) in the revised manuscript). In the revised form of the paper, we have produced a more contextualized introduction to the research investigation.
Reviewer 4 Report
The authors develop a sandwich format electrochemical DNA biosensor and compare GCE with CNT-coated CGE substrates. The topic is interesting but the publication of this study is premature. A lot of work is still needed to make a solid paper. Some comments:
Literature review is far from complete. Some suggestions after a brief search: classic papers: 10.1021/ja053094r, 10.1021/nl060613v, 10.1021/nl051261f, more recent work: 10.1016/j.snb.2017.08.164, 10.1016/j.snb.2017.04.080
Fig. 2 shows a comparsion of CGE and CNT. It should be shown for the same DNA concentrations!
Nanomolar detection limits are not very impressive. What are the required detection limits for the applications the authors have in mind?
Controls are completely missing! The authors should show response to mismatched target DNA, response without capture probe, response without signalling probe. It is impossible to judge whether the sensor is selective.
Repeatability and stability should be addressed as well. How stable is the signal if the measurement is repeated? How much is the drift? How much does the response vary from chip to chip?
The context is missing. Is this sensor any better than state of the art? The authors should compare the sensor at least with other CNT sensors and other sandwich assay for DNA detection.
I recommend rejection of the manuscript in its current form.
Author Response
The authors develop a sandwich format electrochemical DNA biosensor and compare GCE with CNT-coated CGE substrates. The topic is interesting but the publication of this study is premature. A lot of work is still needed to make a solid paper. Some comments:
Literature review is far from complete. Some suggestions after a brief search: classic papers: 10.1021/ja053094r, 10.1021/nl060613v, 10.1021/nl051261f, more recent work: 10.1016/j.snb.2017.08.164, 10.1016/j.snb.2017.04.080
Answer R4.1: We thank the reviewer for this insightful comment which we feel have improved our paper. The suggested references have been included in the introduction of the revised manuscript (Refs. 4-8)
Fig. 2 shows a comparsion of CGE and CNT. It should be shown for the same DNA concentrations!
Answer R4.2: Thank you for pointing out this misunderstanding. The experiments reported in Figure 2 were carried out at target DNA concentration equimolar to SP (50 nM), both on SWCNT-SPEs and GC-SPEs. We apologize for not giving this information in the original manuscript. The target DNA concentration was specified at lines 262-263 (section 3.1) and in the caption of Fig. 2 of the revised manuscript.
Nanomolar detection limits are not very impressive. What are the required detection limits for the applications the authors have in mind?
Answer R4.3: This is a good point. Our study is part of a research program dealing with the development of smart and portable sensing devices for food authenticity assessment. As now clarified in the introduction of the revised manuscript, the selected target DNA is a 45-mer sequence specific of genetically modified soy (Roundup Ready soybean). The aim of this paper, which was proposed for the special issue “Nanostructured Surfaces in Sensing Systems”, is the assessment of the performance achievable combining the enhancing properties of single walled carbon nanotubes with the efficiency of PNA probes covalently bound to the nanostructured substrate with respect to corresponding sensors implemented on non-nanostructured glassy carbon electrodes. The present paper does not explore aspects related to the analysis of real samples, since the validation of genosensor based on CNTs for the determination of non-amplified genomic DNA extracted from flours of genetically modified or wildtype soybean was dealt in another paper to be published in Biosensors and Bioelectronics, already accepted. The findings of that study highlight the good performance reached in terms of femtomolar detection limit referred to genomic DNA, allowing the genosensor to significantly discriminate GM from wildtype soy at concentration levels matching the threshold limit (0,9%) fixed by EC Regulation No. 1829/2003 for labelling of foods containing genetically modified materials. A sentence regarding that article has been added in the Introduction (lines 77-80) of the revised manuscript as well as the corresponding reference (22).
Controls are completely missing! The authors should show response to mismatched target DNA, response without capture probe, response without signalling probe. It is impossible to judge whether the sensor is selective.
Answer R4.4: Even this aspect was faced in the other paper dealing with validation of the CNTs-based genosensor for determination of genomic DNA extracted from real samples (see the answer to the previous point). Selectivity was assessed comparing the responses obtained using Full Match (FM) sequence DNA with two targets containing different mismatches on the CP-complementary portion, i.e. a single mismatch (1-MM) and totally scrambled (SCR) DNA sequences. The signals obtained were compared with that of complementary target FM DNA at the same concentration (10 nM) for both sequences, showing excellent selectivity, as evidenced by a signal reduction by 28% in the case of 1-MM sequence and 98% for the SCR sequence.
The findings of the experiments aimed to assess genosensor selectivity were not reported in the present manuscript in order to avoid self-plagiarism, but are confidentially attached for reviewer’s evaluation.
As for the responses obtained in the absence of CP and SP, we carried out experiments showing not significant signals without SP, whereas in the absence of CP the background signals resulted to be comparable with the responses observed in the absence of target DNA (see Fig. 2). In the last case non-specific absorption phenomena were efficiently prevented by use of pyrene as backfilling agent (see sections 2.4.2 and 3.1).
Repeatability and stability should be addressed as well. How stable is the signal if the measurement is repeated? How much is the drift? How much does the response vary from chip to chip?
Answer R4.5: Since the sensors under investigation are single use disposable devices, all the reported mean values and standard deviations were assessed from three independent assays, which were carried out on different SPEs, as clarified in line 152 (section 2.4) of the revised manuscript. As for repeatability of the response from different chips, relative standard deviations (RSD) always lower than 10% were obtained, as stated in section 3.2, line 285.
The context is missing. Is this sensor any better than state of the art? The authors should compare the sensor at least with other CNT sensors and other sandwich assay for DNA detection.
Answer R4.6: At the best of our knowledge we did not find papers in which PNA probes were combined with SWCNTS through covalent immobilization in order to exploit the peculiar features of carbon nanotubes with the efficiency of PNAs for the development of amperometric genosensors.

Round 2
Reviewer 3 Report
The manuscript has been significantly improved and can be published.
Reviewer 4 Report
The authors have addressed my comments. The revised manuscript may be published in this journal.